# Reforming the registration policy of female sex workers in Senegal? Evidence from a discrete choice experiment

**Sandie Szawlowski[1], Carole Treibich[2], Mylene Lagarde[3], El Hadj Mbaye[4], Khady Gueye[4], Cheikh Tidiane Ndour[4], Aurélia Lépine[1]***

**1** Institute for Global Health, University College London, London, United Kingdom, **2** CNRS, INRA, Grenoble INP, GAEL, Univ. Grenoble Alpes, Grenoble, France, **3** Department of Health Policy, London School of Economics and Political Science, London, United Kingdom, **4** Gouvernement du Sénégal, Ministère de la Santé et de la Prévention Dakar, Dakar, Sénégal

* a.lepine@ucl.ac.uk

## Abstract

Evidence suggests that treating sexually transmitted infections (STIs) amongst female sex workers (FSWs) is a cost-effective strategy to reduce the spread of HIV/AIDS. Senegal is the only African country where sex work is regulated by a public health policy which aims to monitor and routinely treat STIs. The law requires FSWs to be at least 21 years old, register with a health centre and the police, carry an up-to-date registration booklet, attend monthly health check-ups, and test negative for STIs. Despite health and legal benefits of registration, 80% of FSWs in Senegal are not registered. Hence, the potential health benefits of the policy have not materialised. To understand why FSWs do not want to register and to define policy changes that would increase the registration rate of FSWs in Senegal, we designed and implemented a discrete choice experiment (DCE) completed by 241 registered and 273 non-registered FSWs. Participants made choices between a series of hypothetical but realistic registration policy changes. Conditional logit models were used to analyse the DCE data. The results highlighted that confidentiality at the health facility was an important element, registered and non-registered FWs were respectively 26.0 percentage points (pp) and 22.1 pp more likely to prefer a policy that guaranteed confidentiality at the health centre. Similarly, both groups preferred a policy where their health record was only held at the health centre and not with the police. Several interventions to increase FSW registration rate and improve their wellbeing may be implemented without modifying the law. For example, the introduction of psychosocial support in the registration policy package, replacing the registration booklet by a QR code, the use of electronic medical files and the integration of FSWs routine visits with maternal health appointments to increase confidentiality have the potential to encourage registration of FSWs.

**Data Availability Statement:** All relevant data are within the manuscript and its Supporting Information files.

**Funding:** We have now corrected this to MRC Public Health Intervention Development Scheme (MR/T00262X/1). The study was funding by the Medical Research Council, a UK research council under the competitive scheme Public Health Intervention Development Scheme. The funders had no role in study design, data collection and analysis, decision to publish, or preparation of the manuscript.

**Competing interests:** The authors have declared that no competing interests exist.

## Introduction

Globally, there is strong evidence that eliminating sexually transmitted infections (STIs) among female sex workers (FSWs) is a cost-effective strategy to reduce the spread of HIV/AIDS [1]. In West Africa, FSWs are one the key groups disproportionally affected with HIV, with 75% of HIV infections among men directly attributable to sexual intercourse with FSWs [2, 3]. In Senegal, the prevalence of HIV is 6.6% and FSWs are up to 9 times more likely to be infected with HIV than the general population [4]. While most countries have acknowledged sex work as a public health concern, only a few have used registration and monitoring STIs in FSWs as a policy to control the spread of HIV/AIDS in the general population. Senegal is currently the only African country that regulates sex work with a public health policy [5, 6].

The registration policy originates in old French colonial laws [6]. In the second half of the 19th century, the devastating consequences of syphilis, especially in French colonial towns and among troops, was a real concern for the colonial authorities. They found that the hospitalization of soldiers compromised the performance of the military. At the time, medical authorities determined that sex work was at the origin of syphilis transmission. To respond, French legislation introduced the regulation of sex work to prevent the spread of STIs in colonies. In 1849, a decree introduced compulsory medical visits for sex workers and required them to register with the health authorities and carry a health booklet [7, 8]. Sex work was outlawed by the colonial authorities in 1946 and many Francophone countries dropped regulation of sex work after independence. However, Senegal did not. The Senegalese Minister of the Interior at the time decided that regulating sex work was safer than outlawing it to limit its expansion [9]. As a result, since February 1966, Senegalese FSWs older than 21 years old have been required to register with a health centre and attend routine health visits to be tested and treated for STIs [10]. Upon registration, a registration file is created at the health centre and shared with the police. In addition, an official registration booklet is issued (called "*carnet sanitaire*") which keeps a record of the visits made to the appointed health centre. If FSWs test positive for any STI, apart from HIV, the booklet is kept at the health centre during the whole course of treatment. Screened HIV-positive FSWs are referred to higher levels of care, and adherence to antiretroviral treatment is monitored during routine visits. FSWs who are arrested and fail to present an up-to-date registration booklet (either because they are not registered, do not comply with routine visits, or are currently being treated for an STI) may incur a prison sentence of between 2 and 6 months (Code pénal articles 319/ 325, law n° 66–21). Legally, when a FSW chooses to leave sex work she must notify both the health centre and the police, and the registration file held by both authorities should be destroyed. However, in actuality, due to the common belief that FSWs are likely to return to sex work and the want to minimise administrative procedures, the destruction of records are not consistently practiced [10]. Since the introduction of the registration policy no significant change has been implemented, except for minor adjustments to the appearance of the registration booklet.

Despite the potential benefits of the registration policy, it is estimated that 80% of FSWs in Senegal and 57% in the capital city, Dakar, are not registered [4, 11]. To further investigate this finding, in 2015 we established a cohort study of 600 FSWs (300 registered and 300 non-registered FSWs) in Dakar, Senegal which led to the first evaluation of the effect of the registration policy [12]. The study concluded that while FSWs' registration decreased STIs prevalence by 38%, two key problems have limited its benefits. Firstly, registration is unattractive to FSWs as the downfalls of registering outweigh the health and legal benefits to a FSW. Secondly, registration detrimentally impacted FSWs' mental health and well-being, which can lead to riskier health behaviours. These problems were driven by the belief that registration increases the probability that their sex work activity will be discovered by others. If working legally as a

registered FSW, they need to carry and hide their registration booklet while at home, meet clients in public places and attend compulsory health clinics on a specific day (Thursdays), where other patients may realise the reason of their appointment on that specific day. Finally, their registration file and hence, personal information is often kept indefinitely on police and health centre records even if they leave sex work. Despite its legal status, the study found that sex work is morally condemned by society in Senegal and the fear of becoming a social outcast if their activity is discovered is a significant barrier to registration. This finding is also supported by another study which found that sex workers face high levels of social stigma [11]. Key changes to the current policy design are therefore needed to improve its effectiveness and further reduce the spread of STIs among FSWs and in the general population.

This study aimed to identify which aspects of the registration policy could be altered to make it more attractive to FSWs. The remainder of the paper is organized as follows. In Section 2, we present our methodology. Section 3 presents the results and section 4 discusses the findings. Finally, Section 5 concludes.

## Methods

A Discrete Choice Experiment (DCE) was implemented which allowed us to: (1) estimate the elements of a policy that are attractive or unattractive to FSWs, and (2) predict the registration uptake for the policy that combines the most attractive components. To do this, participants were required to make choices between a series of hypothetical, experimentally designed, but realistic registration policies, differing the characteristics of the registration policy. The DCE enabled the identification of policy attributes both deterring FSWs from registering and those that may persuade them to register.

### Data collection and ethics

The DCE was embedded within the third wave of data collection of a longitudinal cohort study of 600 FSWs in Dakar, Senegal, noted earlier, and was carried out from 29 June to 28 July 2020. Data was collected face-to-face by trained enumerators in private rooms located near four of out five of the STI health centres in Dakar, Senegal (Pikine, Rufisque, Mbao, and Sebikotane). These STI health centres are now referred to as the study sites.

The cohort study follows FSWs in Dakar, Senegal and it was conceived to document and understand the socio-economic and behavioural determinants of risky sexual behaviours among FSWs. The cohort study began in 2015 (wave 1) and wave 2 was conducted in 2017. The aim of the third wave of data collection in 2020 was to define a policy change to address the main barriers of registration, increase the benefits of registration, provide evidence on the feasibility and to evaluate the effectiveness and cost-effectiveness of this policy change. Each wave of data collection for the cohort study received appropriate individual ethical clearance. Wave 3 received approval from the National Ethics Committee in Senegal (SEN19/88 n[0]0037MSAS/DGS/DLM/DLSI) and the University College London ethics committee (17341/001).

### Recruitment

For the first wave in 2015, registered FSWs were recruited by the midwife in charge of their compulsory monthly health check. All active registered FSWs from the four study sites were contacted to participate in the study. There was no sampling process employed at the health-centres. Non-registered FSWs were contacted and recruited by leaders of FSW groups. Again, there was no specific process used by leaders of the FSW groups. The total study sample

included a similar proportion of registered and non-registered FSWs. All participants gave permission to be recontacted for future waves.

The same methodology was used to recruit participants in wave 2 (2017) and wave 3 (2020). For wave 2, all participants in wave 1 were contacted to participate. For wave 3, all participants that participated in wave 2 were contacted alongside those that participated in wave 1 but did not participate in wave 2. Participants were initially contacted via telephone and if not contactable, peer FSWs reached out to non-registered sex workers and midwives approached registered FSWs. For both wave 2 and 3, to maintain a cohort of 600 FSWs new registered were recruited at STI health centres and non-registered FSWs were recruited by leaders of FSWs groups. Only participants 18 years old and over and living in Dakar were invited to take part in the study, no minors were included.

## Survey

In wave 3, as previously noted, the survey was conducted in private rooms located near the study site STI health centres in Dakar and strict confidentiality was kept. In wave 1 (2015) and wave 2 (2017) the surveys were conducted face-to-face in the health centres however, due to the COVID-19 the survey locations for wave 3 were relocated to minimise the risk of COVID-19 infection. For all waves, if interested in participating, potential participants were invited to come to one of the four named sites and informed consent followed the same process. Consent was in-person and written. Before consenting to take part in the study, the research purpose, processes and what was expected of them were explained to them. In addition, potential participants were made aware of data management procedures and that all personal information collected during the study will be de-identified to ensure its confidentiality. Respondents were informed that they could withdraw from the study at any time. When ready to consent, all respondents signed a form agreeing to take part in the study.

Survey participants were reimbursed CFAF 2,000 (~£3) for their transport costs and the time spent at the health facility. The full survey was administered face-to-face in Wolof, by local and experienced enumerators, on electronic tablets and took on average 1.5 hours to complete. The DCE was administered early on to ensure that participants were not suffering from survey fatigue and took on average 10 minutes to complete.

Non-anonymized data were sent to a secure server hosted at UCL and were anonymized for analysis.

## Selection and development of attributes and levels for the DCE

The attributes included in the DCE were chosen using findings from previous research conducted on the registration policy in Senegal [12] and discussions with policy experts, government officials, FSW group leaders, and FSWs (registered and non-registered). Previous research found that FSWs have a low level of well-being due to the stigma attached to the current design of the registration policy [12] and the further discussions allowed the identification of key policy attributes in a FSWs registration decision which are feasible to change in legislation.

**Focus groups with female sex workers.** In wave 3 of the data collection there were four focus groups, two with registered FSWs and two with non-registered FSW. Focus groups were conducted in a 'non-directive' approach, facilitated by a topic guide to understand their views on the registration policy and lasted on average 90 minutes. FSWs recruited for the focus group discussions had participated in at least one previous wave of the cohort study and were called at random. Four focus groups (two for registered and two for non-registered FSWs) were conducted with six FSWs in each focus group. The focus groups founded that nine out of

11 (81.8%) non-registered noted that they did not want their profession to be discovered by their loved ones. The findings from the focus groups regarding the favoured interventions to improve the current registration policy among non-registered FSWs along with the methods used and sample size can be found in the supplementary files, S1 and S2 Tables respectively.

**DCE attributes and levels.** The attributes covered key policy attributes of the registration policy and are presented in Fig 1 (column 1). The chosen attributes include the registration file, the design of the registration proof, confidentiality of routine visits, costs of routine visits and psychological support.

The first attribute shown in Fig 1 is the location, or with which authority, a FSWs file is held and for how long their file remains active. According to the law, at the time of registration at the health centre a file is created. This file is then shared with the police. The lawful process for when a FSW decides to leave sex work is that she should notify the health centre and then both the health centre and the police should destroy her file. However, in practice, the location of where and the length of time that the file is held for is often not in keeping with regulation. Files collated at the health centre are not always shared with the police and because of the perceived likelihood that they will return to sex work, health centres do not consistently destroy files when a FSW notifies them that they have left sex work. Discussions with FSW group leaders and FSWs highlighted that having sex work files held by police for life may cause difficulties due to stigma by the police.

The second attribute was the design of the registration proof. To work legally, registered FSWs must carry an up-to-date registration booklet whilst soliciting clients which when not working, they must hide at home. The current booklet easily identifies an individual as a FSW and poses an increased risk of others discovering her sex work activities. As noted, previous research using earlier waves of data found that the booklet was one of the main reasons for non-registration [12]. To respond to this finding, the Ministry of Health in Senegal have attempted changed the appearance of the registration booklet to make it more discreet and less stigmatizing, however there has been no impact on the registration rate [12].

Thirdly, confidentiality of routine visits, was highlighted as an important attribute. Compulsory health visits, outlined in law, for FSWs are held on a specific day (Thursday) which the general population are aware of [5]. FSWs are often forced into attending health centres in their local area due to financial and time constraints. Attending health centre visits locally increases the likelihood of a FSW seeing someone they know and having their sex work activities discovered by others. The focus groups found that 16% of non-registered FSWs believe that medical visits would not be confidential, whereas 98% of FSWs who are registered declare that confidentially is guaranteed. This attribute captures any perceived or real risks of having the sex worker status disclosed whilst at the hospital, either because the hospital staff reveals that the patient is a FSW or if the services for FSWs are not integrated and the FSWs meets someone she knows while queuing or receiving the medical services.

The cost of visit was included as the fourth attribute in the DCE. Although not the most important reason for non-registration in the focus groups, some FSWs expressed that they were unable to meet this expense. FSWs must pay for the compulsory appointment, all STI tests and any treatment. On average, FSWs pay 2000 FCFA (2.63 GBP) per visit. The amount that a FSW pays per visit is not determined by law and varies from health centre to health centre.

The focus groups founded that psychological support was a preferred intervention to increase the likelihood of registration. This fifth attribute is especially important as findings suggest that registering reduces well-being [12].

For each attribute several levels were defined, these are presented in column (2). The levels were determined with the support of policy experts, government officials, FSW group leaders, and FSWs respectively to ensure the feasibility of policy change and suitability.

| (1) Attributes | (2) Levels | (3) Graphical representation of levels |
|---|---|---|
| File | • File at the health centre whilst a FSW<br>• File at the health centre for life<br>• File at the health centre and with police whilst a FSW<br>• File at the health centre and with the police for life | |
| Registration proof | • Current booklet<br>• Current booklet without the words "carte sanitaire"<br>• Smaller version of the current booklet<br>• QR code | |
| Confidentiality at the health centre | • Confidentiality guaranteed<br>• Confidentiality not guaranteed | |
| Cost of routine visits | • Paid (2 000 FCFA)<br>• Free | |
| Psychological support | • No psychological support<br>• Psychological support | |

**Fig 1. Attributes, levels, and graphics included in the discrete choice experiment.**

## Experimental design

The five chosen attributes and their corresponding levels (shown in Fig 1) have a possible $4^2 2^3$ = 48 different combinations of outcome scenarios (two attributes with four levels and three attributes with two levels). All potential scenarios were not presented to each respondent due to likely respondent fatigue and low response rates. Using Ngene 1.2 software [13], a fractional factorial experimental design was used to reduce the number of scenarios whilst maximising the variation in the data. An efficient design was used, allowing for attributes to be independently varied over scenarios while minimising the standard errors of the parameter estimates. Specifically, we used a D-efficient design in which the D-error is minimised [13]. The final optimal design included 20 choice pairs. To reduce cognitive burden and fatigue for the respondents, these 20 choice pairs were 'blocked' and allocated across two versions of the DCE questionnaire each with ten choice pairs.

Allocation to one of the two versions of the DCE is stratified by registration status (i.e., registered and non-registered) and then randomly allocated within each group. The two blocks of policy scenarios were the same for both registered and non-registered FSWs. This was to ensure the successful recommendation of an aligned registration policy which satisfies both types of FSWs.

The attributes and levels presented in the DCE were not discussed in-depth with the participants. Each policy choice pair was read out loud to the participant whilst showing the participant the associated diagram (see Fig 1, column 3 for the graphical representations). Negligible discussion defining each attribute was practiced by all enumerators to ensure the elicitation of truthful responses and minimise salience. The DCE manuscript can be found in the S1 File.

Participants were presented with the ten defined choice pairs consisting of two policies, labelled 'Policy A' and 'Policy B'. Participants were asked to choose their preferred policy out of 'Policy A' and 'Policy B' presented, or if they preferred their current situation (e.g., current registration policy or to be non-registered) they could opt-out. Despite participants falling into two distinct categories, practicing legally or illegally, the baseline reference levels of the policy attributes of the DCE varied between participants within groups. Hence, the characteristics of the opt-out option varied between individuals. For registered FSWs, both misunderstanding and inconsistent application of the law played a huge role in their point of reference. For example, as previously discussed, where and how long a FSWs file is held often does not conform to regulation. To account for the misalignment of the baseline level of attributes, information regarding their current situation was gathered before the first task of the DCE (see Table 2). This information would allow for the appropriate analysis of the opt-out results. See Table 2 for participants baseline DCE attributes descriptive statistics. If a participant chose to opt-out, the next question forced the participant to make a choice between 'Policy A' and 'Policy B'.

**Experimental design testing.** The survey instrument was rigorously pretested at the design stage to verify the appropriateness of the precise wording and framing of the attributes and their corresponding levels. This included the testing of the pictures that illustrated each attribute and their corresponding levels as shown in column (3) of Fig 1. The DCE was piloted in two phases, initially to 10 participants. The first version was well understood by participants, so no changes were made. However, we piloted the DCE on an additional 7 participants in order to collect enough data to use as priors to generate the final efficient experimental design using Ngene 1.2 software [13]. Unexpected priors were highlighted during this stage and qualitative research was undertaken to shed light on them. For example, some participants who were not registered had a registration booklet. Participants were found to be sharing registration booklets and using out-of-date registration booklets to reduce the risk of facing imprisonment when working as a non-registered FSW.

## Analysis

The analysis of the DCE was conducted by estimating a conditional logit model (i.e., a multinomial model) using Stata version 15.0 [14]. The model assumes that all attributes have an independent influence on FSWs preferences and individual $i$ is expected to make choices such that they maximise utility over the policy attributes, shown in Fig 1. The following equation was estimated:

$$U_{ij} = \alpha + \beta' X_{ij} + \varepsilon_{ij} \tag{1}$$

where $U_{ij}$ is the indirect utility function of an individual $i$ for alternative $j$, $\alpha$ is the alternative-specific constant term (for choosing policy A), $\beta' X_{ij}$ is the vector of preferences for the attributes and associated levels included in the DCE (see Table 1), and $\varepsilon_{ij}$ the random component (unobservable variation).

The DCE data contains 20 observations from the 10 choice pairs per survey respondent. Each observation is one of the two alternatives from the 10 choice pairs presented, and with the dependent variable equal to one or zero for each choice pair. In the estimation of the model, all variables were coded as dummy variables. The omitted reference category for each attribute and associated levels are as follows: file–health and police file for life; proof of registration–current booklet; confidentiality at the health centre–confidentiality not guaranteed; cost of medical visits and tests–paid (2 000 FCFA) and psychological support–no psychological support.

**Table 1. Respondent demographic and sex work characteristics.**

| Variables | Registered FSWs | | Non-registered FSWs | |
|---|---|---|---|---|
| | N | Mean/No. | N | Mean/No. |
| *Socio-demographic characteristics* | | | | |
| Age, year (IQR) | 241 | 39.44 (32–47) | 273 | 38.60 (30–47) |
| Divorced or separated (%) | 241 | 181 (75.1) | 273 | 181 (66.30) |
| Working in sex work, months (IQR) | 241 | 99.48 (48–120) | 273 | 84.28 (36–120) |
| Went to school (%) | 241 | 117 (48.55) | 273 | 147 (53.85) |
| Household size (IQR) | 241 | 5.36 (1–7) | 273 | 8.33 (5–11) |
| Risk preferences in general (1 to 10) (IQR) | 241 | 3.89 (2–5) | 273 | 4.08 (3–5) |
| Risk preferences in sex (1 to 10) (IQR) | 241 | 1.96 (0–3) | 273 | 2.19 (0–3) |
| Life satisfaction (1 to 5) IQR | 241 | 3.07 (2–4) | 273 | 3.24 (3–4) |
| *Sex work activity* | | | | |
| Monthly sex work income (CFAF) | 238 | 159,987.40 (60,000–200,000) | 273 | 101,754.60 (50,000–1,200,000) |
| Work mostly in bars or brothels (%) | 241 | 84 (34.85) | 273 | 32 (11.72) |
| Work mostly at home (%) | 241 | 67 (27.80) | 273 | 109 (39.93) |
| Has only occasional clients (%) | 241 | 11 (4.56) | 273 | 7 (2.56) |
| Has only regular clients (%) | 241 | 74 (30.71) | 273 | 266 (97.44) |
| Last client was a regular client (%) | 241 | 170 (70.54) | 273 | 245 (89.74) |
| Declared use of condom with last client (%) | 241 | 235 (97.51) | 273 | 263 (96.34) |
| Number of clients within a week (IQR) | 241 | 6.30 (0–9) | 273 | 4.79 (0–6) |
| Price of last sex act (CFAF) | 241 | 153,44.40 (5,000–18,000) | 273 | 14,909.96 (5,000–20,000) |
| *Link with the authorities and the health system* | | | | |
| Police violence in the last 12 months (%) | 241 | 20 (8.30) | 273 | 10 (3.66) |

Notes: N stands for the number of observations; CFAF: CFA francs

## Results

Among the 604 FSWs who took part in the longitudinal cohort study, 90 were not currently working in sex work. The DCE was only completed by FSWs still in sex work. 273 non-registered and 241 registered FSWs completed the DCE. Demographic and sex work characteristics for both registered and non-registered FSWs that completed the DCE are summarised in Table 1. On average FSWs were 39 years old and 66.3% and 75% of non-registered and registered FSWs, respectively, were divorced or separated. A non-registered FSWs has worked in sex work for an average of 84 months whereas, on average, a registered FSW has worked in sex work for slightly longer, 99 months. With regards to their sex work activity, registered FSWs on average earn more than non-registered FSWs per month (159,987.40 FCFA vs 101,754.60 FCFA). The majority of unregistered FSWs only have regular clients (97.44%), whereas only 30.71% of registered FSWs only have regular clients. In addition, registered FSWs are more likely to solicit in bars or at a brothel compared to their unregistered counterparts who work mostly at home.

Table 2 also presents FSW DCE attribute baseline descriptive statistics. As noted, the baseline of participants within groups varied due to the heterogeneity in perceptions and experiences of FSWs regarding the current law founded by the lack of consistency of the application of the law. Hence, when participating in the DCE it was important that the participant used her perception of her current situation. Nearly 6% of registered FSWs believed that they had no file (at both the health and police level) despite this being a legal requirement and over 2% of non-registered FSWs believed that they had a health file held by the authorities. It was founded that a higher proportion of non-registered FSWs received free care (69.12%) compared to registered FSWs (42.80%). In addition, we found that half of the participants benefited from some form of psychosocial support. However, from further investigation it was

**Table 2. Participant baseline levels of DCE attributes and opt-out statistics.**

|  | Registered FSWs | | Non-registered FSWs | |
|---|---|---|---|---|
|  | N | Mean/No. | N | Mean/No. |
| **DCE baseline levels for participants** |  |  |  |  |
| File |  |  |  |  |
| No file (%) | 241 | 14 (5.81) | 271 | 265 (97.79) |
| File at health centre (%) | 241 | 149 (61.83) | 271 | 5 (1.85) |
| File at health centre and police (%) | 241 | 78 (32.37) | 271 | 1 (0.37) |
| Length of time of file |  |  |  |  |
| *File with health centre* |  |  |  |  |
| File at health centre whilst FSW (%) | 142 | 136 (95.77) | 3 | 3 (100.00) |
| File at health centre for life (%) | 142 | 6 (4.23) | 3 | 0 (0.00) |
| *File with police* |  |  |  |  |
| File with police whilst FSW (%) | 74 | 68 (91.89) | 1 | 0 (0.00) |
| File with police for life (%) | 74 | 6 (8.11) | 1 | 1 (100.00) |
| Does not have a 'carte sanitaire' (%) | 241 | 9 (3.73) | 272 | 263 (96.69) |
| Believe that health centres are confidential (%) | 240 | 235 (97.92) | 248 | 209 (84.27) |
| Attends health centre visits (%) | 241 | 237 (98.34) | 272 | 205 (75.37) |
| Receives free care during health centre visits (%) | 236 | 101 (42.80) | 204 | 141 (69.12) |
| Receives psychological support (%) | 241 | 146 (60.58) | 269 | 108 (40.15) |
| **DCE opt-out** |  |  |  |  |
| Opt-out at least once (%) | 241 | 21 (8.71) | 273 | 43 (15.75) |
| Opt-out rate of all choices (%) | 2410 | 32 (1.33) | 2730 | 233 (8.53) |

Notes: N stands for the number of observations. FSW: Female sex worker.

noted that this support was informally delivered by untrained peer sex workers and hospital staff which is likely to have hampered the transparency and efficacy of such support.

The two opt-out statistics presented in Table 2 are (1) the percentage of participants that opted-out of making a choice between Policy A and Policy B at least once and (2) the percentage of participants that opted-out for each one of the 10 policy scenarios presented. 43 non-registered FSWs out of 273 (15.75%) and 21 registered FSWs out of 241 (8.71%) chose the opt-out option at least once when presented with the 10 policy scenarios. The opt-out rate of all choices (both registered and non-registered FSWs) was very low 5.16% (265 out of 5140 choices), 1.33% (32 out of 2410 choices) for registered FSWs and 8.53% (233 out of 2730 choices) for non-registered FSWs.

To analyse the DCE, we estimated two separate conditional logit models for each sub-sample of FSWs (registered and non-registered). The absolute values of the coefficients (preference weights) do not have any meaningful interpretation as they measure relative preferences. The average marginal effects of each attribute, which determines the changes between attribute-level estimates and the relative sizes of those changes across attributes, were calculated to estimate how the likelihood to prefer a certain policy varies depending on attribute changes. Given the low rate of opt-out we did not analyse this option and only analysed the 'forced' choice set. This does not cause any issues with the data or the analysis. The findings are shown in Table 3.

Table 3 reports the marginal effects of a policy attribute on the probability that respondents would prefer this policy. The results are shown separately for registered and non-registered FSWs. For both groups, confidentiality guaranteed at the health centre level is the most important policy attribute. Registered FSWs are 26.0 percentage points more likely to prefer a policy that guarantees their anonymity, compared to a situation where registration does not guarantee confidentiality at the health centre. Similarly for non-registered FSW confidentiality is more likely to be preferred by 22.1 percentage points.

The next most important aspect also relates to issues of confidentiality, since it is where a file on a FSW is held, and for how long. Registered and non-registered FWs are respectively 14.6 and 17.0 percentage points more likely to prefer a policy where their health file is only held at the health centre level, compared to the current situation where their file held at both a health centre and by the police for life.

For proof of registration, compared to the current 'carnet sanitaire' the adoption of a QR code is shown to improve probability of preferring a policy by 5.9 (registered FSWs) and 3.5 (non-registered FSW) percentage points.

Psychological support is also seen to be important, with the provision of support improving the likelihood of preferring a policy by registered and non-registered FSWs by 5.1 and 5.7 percentage points, respectively.

Finally, the cost of health centre visits and tests are found to not be an important attribute in the registration decisions of both registered and non-registered FSWs i.e., there is no significant difference in preference between paid and non-paid health visits.

Hence, according to FSWs, ensuring confidentiality at health centre, replacing the physical registration proof by a QR code, providing psychological support, removing the registration file held by the police and only holding the registration file at the health centre whilst active in sex work would significantly improve the registration policy.

## Discussion

Senegal is the only African country where sex work is legalised and regulated, and this study is the first to use a DCE to elicit the registration policy preferences of FSWs in Africa. The

**Table 3. FSW policy preferences.**

| | Average Marginal effects | |
| --- | --- | --- |
| | **Registered FSWs** | **Non-registered FSWs** |
| **File (ref = File at health centre and police for life)** | | |
| File at health centre whilst a FSW | 0.146*** | 0.170*** |
| | [0.119–0.174] | [0.146–0.195] |
| File at health centre for life | 0.030*** | 0.014 |
| | [0.011–0.050] | [-0.004–0.032] |
| File at health centre and police whilst a FSW | 0.082*** | 0.106*** |
| | [0.055–0.109] | [0.082–0.129] |
| **Proof of registration (ref = Current booklet)** | | |
| Current booklet without the words 'carnet sanitaire' | 0.040*** | 0.033*** |
| | [0.019–0.062] | [0.012–0.053] |
| Smaller version of the current booklet | 0.006 | 0.018 |
| | [-0.016–0.028] | [-0.004–0.040] |
| QR code | 0.059*** | 0.035*** |
| | [0.036–0.082] | [0.015–0.054] |
| **Confidentiality at the heath centre (ref = Confidentiality not guaranteed)** | | |
| Confidentiality guaranteed | 0.260*** | 0.221*** |
| | [0.239–0.282] | [0.202–0.241] |
| **Cost of health centre visits and tests (ref = 2 000 FCFA)** | | |
| Free | -0.007 | 0.003 |
| | [-0.020–0.005] | [-0.008–0.014] |
| **Psychological support (ref = No support)** | | |
| Psychological support | 0.051*** | 0.057*** |
| | [0.031–0.072] | [0.039–0.075] |
| N | 4,794 | 5,340 |
| Clusters in ID | 241 | 270 |
| Prob > chi2 | 0.0000 | 0.0000 |
| Log pseudolikelihood | -1100.9688 | -1381.9941 |
| AIC | 2219.938 | 2781.988 |
| BIC | 2278.214 | 2841.235 |

Coefficients are average marginal effects.

***$p \leq 0.01$

**$p \leq 0.05$

*$p \leq 0.1$

N stands for the number of observations. Standard errors are robust and clustered at the individual level. 95% confidence intervals in square brackets. The marginal effect is calculated by dividing the coefficient of the attribute by the change in the independent variable. The average marginal effects of each attribute were calculated to understand how much the rate of registration changes when an attribute changes. To do this you calculate the marginal effect for each individual i in the sample and then average the marginal effect over all sampled individuals. CFAF: CFA francs. FSW: Female sex worker.

Reference categories are as follows: File–health and police file for life; Proof of registration–current booklet; Confidentiality at the health centre–confidentiality not guaranteed; Cost of medical visits and tests–paid (2 000 FCFA) and; Psychological support–no psychological support.

findings identified registration policy attributes that both deterred FSWs from registering and those that encouraged them to register. Furthermore, to the best of our knowledge there are no other similar studies in non-African countries. While only a few countries globally have implemented legislation which requires FSWs to register and follow compulsory check-ups/ STI testing, our study sheds light on the issues that would arise with such laws. Our results also identified important design aspects to consider when implementing such a policy.

Our study explained why the majority of FSWs in Senegal preferred to practice sex work illegally rather than register as a FSW to practice legally. Moreover, it identified policy changes that would make registration more attractive to FSWs. An important finding from the study is that preferences of registered and non-registered FSWs were aligned, meaning that changes to increase registration would also improve the wellbeing of already registered FSWs.

The DCE found that the most important policy attribute to FSWs was guaranteed confidentiality at the health centre level. Both registered and non-registered FSWs prefer to have a file at the health centre whilst a FSW (as opposed to for life), a QR code as the main form of registration identification instead of the registration booklet, guaranteed confidentiality at the health centre and psychological support, and lastly free health visits was not significantly different.

We found that confidentiality was a key aspect of the willingness to register. Even if many precautions were put in place to maintain the confidentiality of FSWs, there were still some potential issues that could result in a confidentiality breach. Firstly, some health centres still have designated days for the FSWs legally required health visit, which makes them identifiable to the rest of the population. Some health facility staff have also been reported to act differently with FSWs, which could also risk revealing their activity to the public [12]. This could be addressed by making sure that all health centres are incorporating FSWs' visits with the general population or with maternal care services, with no designated days for FSWs routine visits. Also, consistently renewing the training of health staff in keeping confidentiality would support the protection of FSWs. Secondly, the registration booklet FSWs carry around and must present to health staff during their health visit is a form of identification which they are afraid of. If the registration booklet was misplaced, having the picture, name and health information easily accessible puts a huge stress on FSWs, who have to hide their booklets from their friends and family members. Replacing this physical booklet by a QR code would improve confidentiality, as it would not contain a picture or any readable information.

Finally, providing psychological support as part of the registration package was favoured in the DCE results. Many FSWs experience physical and psychological violence from their clients, other FSWs and the police. In addition, many of them experience high stress levels from having to hide their activity from their friends and family members. However, it was founded that the health staff and certain sex work group leaders were already filing the role of psychological counsellor informally despite their lack of time, training, and resources. In Senegal, as per many other sub-Saharan countries, psychologists are rare [15, 16] and their fees are not affordable for FSWs or small Non-Government Organisations (NGOs), reinforcing the need for a state-funded service available to all key populations, and offered at health centre level part of the registration package.

Regarding the feasibility of the implementation of the policy recommendations, discussions during dissemination of results to policy stakeholders revealed that it was impossible to remove or significantly change the current law. Since its introduction, no significant changes to this policy have been implemented, and consequently it is ingrained into the fabric of Senegalese society. Politicians that attempt to remove this law would risk huge reputational damage, upheaval from their position and being outcasted. In addition, such law amendments are unlikely to obtain a majority of votes at the parliament. Given this, it would be more impactful to improve the working conditions of sex workers to attract them to services which can truly benefit and protect them.

Whilst only a few countries have opted for compulsory medical check-ups to treat STIs among FSWs, two literature reviews found that punitive laws against FSWs are counterproductive to HIV prevention efforts [17, 18]. Our results give further evidence suggesting that laws could lead to an increase in stigma and human rights violations, and hence increasing the

likelihood that, already vulnerable, FSWs practice sex work illegally. Urgent policy changes need to be considered by countries regulating sex workers using punitive laws to limit the negative effects of those policies on the mental and physical health of FSWs.

The findings of the study and proposed design of the registration policy focuses on improving FSWs confidentiality and mental health and combating the potential issues of punitive laws. These issues are universally a challenge for FSWs. Hence, if looking to implement laws to regulate sex work via compulsory health check-ups or STI testing, this study is likely to also support the design of an acceptable and feasible policy for sex workers in other settings.

There are several limitations to this study. Firstly, the DCE may lack external validity if FSWs do not make the same choices in real life. Secondly, despite the outcome choices presented in the DCE being realistic and based on real data, the choice task was hypothetical. Thirdly, due to ethical reasons we were not able to include under-18 FSWs. It can therefore be argued that these findings are not generalisable to the whole population of FSWs. Finally, the sample of registered FSWs is likely to be representative of this population. However, because snowball sampling method was adopted to recruit unregistered FSWs it is likely that those closely connected to FSWs groups are oversampled. Nevertheless, changes in policy to encourage registration is unlikely to reach the most isolated unregistered FSWs. Hence, the results from the DCE presented remain valid and likely to be founded if changes to the policy were to be made.

## Conclusion

While FSWs registration has the potential to limit HIV acquisition and transmission, the current issues in the design and implementation of the policy in Senegal explains the low registration rate of FSWs. The study showed that several policy changes have the potential to increase the registration rate of FSWs and improve their wellbeing without overturning the law. For example, changing the registration proof to a QR code and including psychosocial support in the registration support package would increase the benefits of registration. In addition, integrating registered FSWs mandatory sexual health appointments with maternal health appointments and reinforcing training to ensure confidentiality in health centres have the potential to encourage registration of FSWs.

## Supporting information

**S1 Table. Favoured interventions to improve the current registration policy among non-registered FSWs.** The findings from the focus groups regarding the favoured interventions to improve the current registration policy among non-registered FSWs.
(DOCX)

**S2 Table. Focus group methods and study sample sizes.**
(DOCX)

**S1 File. DCE manuscript.** The script used by enumerators to present the DCE to participants.
(DOCX)

**S2 File. Discrete choice experiment data.** The DCE data collected and analysed for this study.
(DTA)

## Author Contributions

**Conceptualization:** Carole Treibich, Mylene Lagarde, Khady Gueye, Cheikh Tidiane Ndour, Aurélia Lépine.

**Data curation:** Khady Gueye, Cheikh Tidiane Ndour, Aurélia Lépine.

**Formal analysis:** Sandie Szawlowski, Aurélia Lépine.

**Funding acquisition:** Aurélia Lépine.

**Investigation:** El Hadj Mbaye, Khady Gueye, Cheikh Tidiane Ndour.

**Methodology:** Aurélia Lépine.

**Project administration:** Sandie Szawlowski, Aurélia Lépine.

**Supervision:** Carole Treibich, Mylene Lagarde, Aurélia Lépine.

**Validation:** Aurélia Lépine.

**Writing – original draft:** Sandie Szawlowski.

**Writing – review & editing:** Carole Treibich, Mylene Lagarde, Aurélia Lépine.

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
