## [Decision Letter · Decision Letter 0]

24 Apr 2023

PONE-D-22-30805Reforming the registration policy of Female Sex Workers in Senegal? Evidence from a discrete choice experimentPLOS ONE

Dear Dr. Lepine,

Thank you for submitting your manuscript to PLOS ONE. After careful consideration, we feel that it has merit but does not fully meet PLOS ONE’s publication criteria as it currently stands. Therefore, we invite you to submit a revised version of the manuscript that addresses the points raised during the review process.

We look forward to receiving your revised manuscript.

Kind regards,

Hamufare Dumisani Dumisani Mugauri, Ph.D. Public Health

Academic Editor

PLOS ONE

“No”

6. Thank you for stating the following in your Competing Interests section: 

“NO authors have competing interests”

7. In your Data Availability statement, you have not specified where the minimal data set underlying the results described in your manuscript can be found. PLOS defines a study's minimal data set as the underlying data used to reach the conclusions drawn in the manuscript and any additional data required to replicate the reported study findings in their entirety. All PLOS journals require that the minimal data set be made fully available. For more information about our data policy, please see http://journals.plos.org/plosone/s/data-availability.

8. We note that you have indicated that data from this study are available upon request. PLOS only allows data to be available upon request if there are legal or ethical restrictions on sharing data publicly. For more information on unacceptable data access restrictions, please see http://journals.plos.org/plosone/s/data-availability#loc-unacceptable-data-access-restrictions.

9. PLOS requires an ORCID iD for the corresponding author in Editorial Manager on papers submitted after December 6th, 2016. Please ensure that you have an ORCID iD and that it is validated in Editorial Manager. To do this, go to ‘Update my Information’ (in the upper left-hand corner of the main menu), and click on the Fetch/Validate link next to the ORCID field. This will take you to the ORCID site and allow you to create a new iD or authenticate a pre-existing iD in Editorial Manager. Please see the following video for instructions on linking an ORCID iD to your Editorial Manager account: https://www.youtube.com/watch?v=_xcclfuvtxQ.

**Comments to the Author**

1. Is the manuscript technically sound, and do the data support the conclusions?

Reviewer #1: Yes

Reviewer #2: Partly

2. Has the statistical analysis been performed appropriately and rigorously? 

Reviewer #1: Yes

Reviewer #2: I Don't Know

3. Have the authors made all data underlying the findings in their manuscript fully available?

Reviewer #1: No

Reviewer #2: No

4. Is the manuscript presented in an intelligible fashion and written in standard English?

Reviewer #1: No

Reviewer #2: Yes

5. Review Comments to the Author

Reviewer #1: ABSTRACT

• The second and third sentence of the abstract seem redundant. Simplify this to include more in the background of the abstract.

• Abstract could use further description of the methods and results of the discrete choice experiment

• Include quantitative results in the abstract

• Use past tense

INTRODUCTION

• There are some important readability concerns in this section, including incomplete/run-on sentences (see paragraph 2). Please review carefully and correct these issues.

• All of the references used are quite old, e.g. cost-effectiveness and importance of addressing the health needs of FSW. Please include more recent and additional citations/evidence to justify your statements.

o There are a number of statements that do not include any reference at all but require them, including most of paragraph 2 and 3.

• You mention that in West Africa, HIV infections are concentrated among FSW. What about other key populations? This statement could benefit from expounding that FSW are one key group that are disproportionately affected by HIV. See PMC4804341.

• I’m not sure what you mean when you say “For medical authorities sex work was at the origin of syphilis transmission.”

• It is unusual to include a detailed summary of the methods and results in the introduction. https://journals.plos.org/plosone/s/submission-guidelines

METHODS

• You mention that you tried to contact FSWs who had taken part in the 2015 and 2017 surveys, but of what? A larger study? An IBBS? A serial cross-sectional study? Please provide more context here

• How was recruitment done? Was there any sampling process employed at the health-centres? Was there a specific process used by leaders of the FSW groups?

• It would be helpful to have more description on the different attributes and how they were determined for inclusion. You mention a literature review – what was reviewed? What were the results of that review that led you to these attributes?

• How were folks selected for the focus group discussions?

• 1st paragraph of pg. 8, there are again some issues with readability

• You indicate that there were two phases of piloting. Is the first the completion of the DCE by 17 participants and the second the analysis to generate priors for the final experimental design? This was unclear.

• Was randomization for the DCE stratified by registration status? It seems that way based on the results but it would be helpful to spell this out

RESULTS

• The table and the discussion of the results is very clear

DISCUSSION

• The discussion is completely lacking references and does not relate the findings of this study to any other literature, quantitative or qualitative.

• There are claims made, such as “In Senegal, psychologists are rare and their fees are not affordable…reinforcing the need for a state-funded service…” but these statements need corresponding citations.

Reviewer #2: This article presents the results of a discrete choice experiment among female sex workers (FSW) in Senegal about their preferences regarding the current registration policy in Senegal. The survey was embedded in a larger longitudinal survey among FSWs. Both registered and non-registered FSWs were surveyed.

The topic is relevant in terms of public health, and the survey results could be used to improve the current policy. However, the paper could benefit from several improvements and clarifications before publication.

The main issue is the way the article is organised. It seems that a previous version of the paper was not following the IMReD format, as suggested by the last paragraph of the introduction: “In Section 2, we present our methodology. Section 3 presents the data and descriptive statistics. Section 4 presents the method to overcome the selection bias associated with the decision to register along with the sensitivity analysis to test the conditional independence assumption. Results and a series of robustness checks are presented in Section 5 and discussed in Section 6. Finally, Section 7 concludes.” The paper has been reorganised to a more classical “Introduction – Method – Results – Discussion” plan, but incompletely, as suggested by this last paragraph of the introduction, not adequately updated.

Some methodological aspects and results are presented in the introduction that should be moved to the appropriate place. Other elements should be clarified (see the following comments). The authors could look at several recent papers from PLoS One as examples for better organising their contents.

I also have an issue with some elements of the conclusion, in particular, that sentence: “This could be addressed by making sure that all health centres are incorporating FSWs’ visits with the general population or with maternal care services.”

The conclusion drawn in the paper appears to be based on discussions held with local stakeholders or qualitative interviews conducted with health professionals, rather than solely on the presented results. However, these elements have not been explicitly presented in the methods or results sections of the paper. It is imperative that the conclusions of the paper are solely based on the presented results. To rectify this, the authors have two options: they may choose to remove this comment from the conclusion altogether or mention it in the discussion section as a potential solution that could be explored. Alternatively, the authors may choose to integrate the results of the workshop with stakeholders into the methods and results sections of the paper. This would ensure that the conclusions drawn are based on a solid foundation of empirical evidence.

INTRODUCTION

“In West Africa, the HIV epidemic is concentrated among FSWs, with 75% of HIV infections among men directly attributable to sexual intercourse with FSWs (Alary & Lowndes, 2004) (Alary M. , et al., 2013)” Do you have any more recent reference? Furthermore, this statement holds less validity for Senegal, where the epidemic is currently being driven by MSM.

The introduction section of the current paper is recommended to be enhanced by incorporating additional references as the existing count of 14 references appears to be insufficient. Furthermore, the inclusion of more recent references is also suggested as only two references have been published within the last five years.

The introduction should first explore West Africa before being more focused on Senegal.

Please provide more context and details about the current policy in terms of FSWs registration, and the current standard of care. About available services? It seems that half of participants already receive psychological support for example.

“registration is unattractive to FSWs, explaining that 80% of FSWs in Senegal (Foley, 2010) and 57% in the capital city” said twice

“This study aimed to identify which aspects of the registration policy could be altered to make it more attractive to FSWs. To do this, a discrete choice experiment (DCE) was administered to 241 registered and 273 non-registered FSWs. Five key policy attributes (registration file, registration identification, health centre confidentiality, health visit costs, psychological support) presented in the DCE were identified through previous research on the registration policy in Senegal and from a qualitative research with registered and non-registered FSWs (Procureur et al, 2022). Conditional logit models were conducted to analyse the DCE data. Results show that registration policy preferences of registered and non-registered FSWs were aligned. According to FSWs, ensuring confidentiality at health centre, replacing the physical registration proof by a QR code, providing psychological support, removing the registration file held by the police and only holding the registration file at the health centre whilst active in sex work would significantly improve the registration policy. As a result, while complete removal of police files and suppression of files held at the health centre when FSWs are no longer active would make registration much more attractive to FSWs, such amendments were deemed unfeasible given the current political context. Nevertheless, other changes potentially effective to increase registration could be introduced without the need to amend the law, including improving FSWs’ confidentiality, providing a psychosocial support service, improving the relationship of FSWs with the police and replacing the current registration card by a QR code to reduce stigma. “ This paragraph presents methods and results and should not be part of the introduction. The different elements should be presented later, at a more appropriate place.

METHODS

“The DCE was embedded within the third wave of data collection of a longitudinal survey” Please provide more information about this longitudinal survey. Does the survey has a name? Has a protocol been published? Other analysis?

Ethics: was it a specific approval for the DCE or a global approval for the longitudinal survey? When was approval obtained? Is there any reference (e.g. registration number) from the local ethic committee?

Text and figure 1 should be harmonised (e.g. order and name of the attributes). Indicate on figure 1 what is the current standard of care. Is there a difference between a card and a booklet? How was the confidentiality at the health centre presented to the participants? How was the question formulated? Is it about perception? What was the exact wording used in the questionnaires?

What is Ngene? A software? Please be explicit and add a reference.

“Specifically, we used a D-efficient design in which the D-error is minimised.” Please explain. Any reference?

“We analysed the data and used the regression coefficients from the analysis of the pilot as priors to generate the final efficient experimental design.” Unclear for me. Could you explain? Clarify?

In terms of data policy, it seems that the data are not available on a data repository.

RESULTS

Table 1: please provide an informative title.

What is the item “File”? Current situation of the participant???

What is Opt-Out in this table? A result of the DCE? Maybe not a good idea to mix some DCE resultst with a description of study participants.

What is “Length of time of file”? Current situation? If so, how do they know that the file will be there for life or just whilst FSW?

Average marginal effects should be explained in the methods, including the note of Table 2. It seems that they are more specifically marginal contrasts, i.e. the difference between marginal predictions. Be aware that there is currently no consensus around the wording “marginal effects” and that it could refer to many things, see for example https://www.andrewheiss.com/blog/2022/05/20/marginalia/

Table 2 requires a more comprehensive and descriptive title to enhance its readability. Additionally, the authors could consider presenting the marginal contrasts in percentage points (as in the text), which would facilitate the interpretation of the results by the readers. The use of standard errors in the table is not convincing as they are not easy to interpret. A more suitable option to display uncertainty would be to use 95% confidence intervals.

To facilitate the reading of the results, the authors could create a new paragraph each time they present a new dimension.

DISCUSSION

Are there similar surveys in the literature? Could the authors compare their results with other studies?

Could results be replaced in context?

6. PLOS authors have the option to publish the peer review history of their article (what does this mean?). If published, this will include your full peer review and any attached files.

Reviewer #1: No

Reviewer #2: **Yes: **Joseph Larmarange

---

## [Author Response · Author response to Decision Letter 0]

17 Jul 2023

Please see our word document with detailed response to each comment made by the reviewer.

---

## [Editor Report · Decision Letter 1]

28 Jul 2023

Reforming the registration policy of Female Sex Workers in Senegal? Evidence from a discrete choice experiment

PONE-D-22-30805R1

Dear Dr. Lépine,

We’re pleased to inform you that your manuscript has been judged scientifically suitable for publication and will be formally accepted for publication once it meets all outstanding technical requirements.

Kind regards,

Hamufare Dumisani Dumisani Mugauri, Ph.D. Public Health

Academic Editor

PLOS ONE

---

## [Editor Report · Acceptance letter]

7 Aug 2023

PONE-D-22-30805R1 

Reforming the registration policy of Female Sex Workers in Senegal? Evidence from a discrete choice experiment 

Dear Dr. Lépine:

I'm pleased to inform you that your manuscript has been deemed suitable for publication in PLOS ONE. Congratulations! Your manuscript is now with our production department. 

Kind regards, 

on behalf of

Mr Hamufare Dumisani Dumisani Mugauri 

Academic Editor

PLOS ONE